# Rapid adaptation to human protein kinase R by a unique genomic rearrangement in rhesus cytomegalovirus

Stephanie J. Child[1], Alexander L. Greninger[2], Adam P. Geballe[1,3]*

1 Divisions of Human Biology and Clinical Research, Fred Hutchinson Cancer Research Center, Seattle, Washington, United States of America, 2 Department of Laboratory Medicine, University of Washington, Seattle, Washington, United States of America, 3 Departments of Medicine and Microbiology, University of Washington, Seattle, Washington, United States of America

* ageballe@fredhutch.org

**Data Availability Statement:** The data are all with in the manuscript file or deposited in the NCBI BioProject database:https://www.ncbi.nlm.nih.gov/bioproject/?term=PRJNA660187.

## Abstract

Cytomegaloviruses (CMVs) are generally unable to cross species barriers, in part because prolonged coevolution with one host species limits their ability to evade restriction factors in other species. However, the limitation in host range is incomplete. For example, rhesus CMV (RhCMV) can replicate in human cells, albeit much less efficiently than in rhesus cells. Previously we reported that the protein kinase R (PKR) antagonist encoded by RhCMV, rTRS1, has limited activity against human PKR but is nonetheless necessary and sufficient to enable RhCMV replication in human fibroblasts (HF). We now show that knockout of PKR in human cells or treatment with the eIF2B agonist ISRIB, which overcomes the translational inhibition resulting from PKR activation, augments RhCMV replication in HF, indicating that human PKR contributes to the inefficiency of RhCMV replication in HF. Serial passage of RhCMV in HF reproducibly selected for viruses with improved ability to replicate in human cells. The evolved viruses contain an inverted duplication of the terminal 6.8 kb of the genome, including rTRS1. The duplication replaces ~11.8 kb just downstream of an internal sequence element, *pac*1-like, which is very similar to the *pac*1 cleavage and packaging signal found near the terminus of the genome. Plaque-purified evolved viruses produced at least twice as much rTRS1 as the parental RhCMV and blocked the PKR pathway more effectively in HF. Southern blots revealed that unlike the parental RhCMV, viruses with the inverted duplication isomerize in a manner similar to HCMV and other herpesviruses that have internal repeat sequences. The apparent ease with which this duplication event occurs raises the possibility that the *pac*1-like site, which is conserved in Old World monkey CMV genomes, may serve a function in facilitating rapid adaptation to evolutionary obstacles.

## Author summary

Rhesus macaque CMV (RhCMV) is an important model for human CMV (HCMV) pathogenesis and vaccine development. Therefore, it is important to understand the similarities and differences in infectivity and interaction of these viruses with their host species.

**Funding:** This work was supported by the National Institute of Allergy and Infectious Diseases of the National Institutes of Health grants NIH R56AI026672 and RO1AI45945 (to A.P.G.) and by the Genomic Core Shared Resource of the Fred Hutch/University of Washington Cancer Consortium (P30 CA015704). The content is solely our responsibility and does not necessarily represent the official views of the National Institutes of Health. The funders had no role in study design, data collection and analysis, decision to publish, or preparation of the manuscript.

**Competing interests:** The authors have declared that no competing interests exist.

In contrast to the strict species-specificity of HCMV, RhCMV is able to cross species barriers to replicate in human cells. We know from past work that a component of this broader host range is RhCMV's ability to counteract both the rhesus and human versions of a key antiviral factor. Here we delve further into the mechanisms by which RhCMV can adapt to counteract human cellular defenses. We find that RhCMV appears to be poised to undergo a specific genomic rearrangement that facilitates increased replication efficiency in human cells. Besides providing insights into CMV species-specificity and host barriers to cross-species transmission, this work also provides more generalized clues about viral adaptative mechanisms.

## Introduction

Cytomegaloviruses (CMVs) diverged from the other herpesvirus subfamilies and have been co-speciating with their host lineages for approximately 60 to 80 million years [1]. The hypothesis that this intimate coevolution has fine-tuned the ability of these pathogens to utilize their host's dependency factors while also evading the host's restriction factors is supported by the observation that many CMVs, including human CMV (HCMV), demonstrate fairly strict species-specificity, with infectivity constrained to only very closely related species [2]. However, several non-human primate CMVs, including rhesus macaque CMV (RhCMV), African green monkey CMV and squirrel monkey CMV are able to cross species barriers and replicate at least to a low level in human cells [3,4].

One host cell restriction factor that has been shown to contribute to species-specificity is protein kinase R (PKR), which is activated by double-stranded RNA (dsRNA), a byproduct of many viral infections including CMV [5,6]. Active PKR phosphorylates the α subunit of eukaryotic initiation factor 2 (eIF2α), causing inhibition of the eIF2 guanine nucleotide exchange factor eIF2B, thereby inhibiting protein synthesis and viral replication [7]. The importance of PKR in the antiviral response is highlighted by the presence of PKR antagonists in many viruses [8]. An evolutionary "arms race" between PKR and its viral antagonists is evident in the strong signature of positive selection in primate PKR genes [9,10]. In many viruses, including CMVs, herpes simplex virus-1 (HSV-1) and vaccinia virus (VACV), deletion of these PKR antagonists severely reduces or eliminates replication in wild type cells and virulence in infected animals [11–15]. However, viral replication and virulence is at least partially restored in PKR-deficient cells and animals [16–21].

Like HSV-1, HSV-2, and New World monkey CMVs, HCMV has a complex genome structure (class E) consisting of a unique long ($U_L$) and a unique short ($U_S$) region, each flanked by inverted repeats [22]. HCMV encodes two PKR antagonists, IRS1 and TRS1, which are ~2/3 identical as they are partially encoded in the repeats surrounding the $U_S$ region of the genome [2]. During replication, CMVs with a class E genome structure generate an equal mixture of four genomic isomers, representing inversions of the $U_L$, $U_S$, or both segments relative to the prototypic orientation. In contrast, Old Word monkey CMVs, including RhCMV, possess a simple genome structure (class A) with one unique region flanked by direct terminal repeats. RhCMV encodes only one known PKR antagonist, rTRS1 (Rh230). We previously showed that although rTRS1 is a very weak antagonist of human PKR, RhCMV is able to replicate in human fibroblasts (HF) and requires rTRS1 to do so [21,23,24]. The levels of rTRS1 produced by RhCMV appear to be sufficient to counteract human PKR enough to allow at least limited RhCMV replication in HF.

Here we have exploited the constraints placed on RhCMV replication by human PKR to determine whether and how RhCMV might adapt to overcome restriction factors such as PKR. Using an experimental evolution strategy, we found that passage of RhCMV in HF reproducibly yielded viruses with increased replication efficiency, and that these viruses express higher levels of rTRS1 due to an inverted duplication of a segment of the genome that includes rTRS1 and the terminal repeat sequence. This duplication event reveals that *pac*1-like, an internal sequence that resembles *pac*1 (the terminal packaging signal), appears to facilitate the genomic rearrangement. Interestingly, as a result of this inverted duplication, the experimentally evolved viral genomes are now capable of isomerization in the same way as herpesviruses with complex class E genome structures. These results demonstrate one means of viral adaptation that could contribute to the cross-species transmission of large DNA viruses. Increasing our understanding of the mechanisms contributing to cross-species transmission of viruses, as highlighted by the current pandemic, is of enormous importance.

## Results

### RhCMV replication in HF is constrained in part by human PKR

Previously we demonstrated that the RhCMV PKR antagonist, rTRS1, is capable of only weakly counteracting human PKR [21,23,24]. However, this limited activity is necessary and sufficient to enable RhCMV to overcome human PKR and replicate in HF. To more carefully evaluate the importance of human PKR as a barrier to RhCMV replication, we infected HF and PKR knock-out HF (HF$^{\Delta PKR}$) with RhCMV and measured the titers of the progeny virus released into the medium over the course of 6 days (Fig 1A). Consistent with previous observations [3,21], RhCMV was able to replicate in HF. However, the virus replicated to significantly higher titers in HF$^{\Delta PKR}$, supporting the conclusion that human PKR restricts RhCMV replication. The finding that RhCMV replicated to even high titers in rhesus fibroblasts (RF) than in HF$^{\Delta PKR}$ suggests that PKR is not the only factor involved in limiting RhCMV infection in human cells.

As a complementary approach, we measured replication of RhCMV in HF in the presence or absence of ISRIB [25], an eIF2B agonist that counters the translational inhibitory effects of eIF2α phosphorylation (Fig 1B). Treatment with ISRIB enhanced RhCMV replication in HF, providing additional support for the conclusion that the PKR pathway in HF restricts RhCMV replication. These results suggested that HFs might provide a useful model for assessing whether and how RhCMV might be able to adapt to replicate better in cells from a non-natural host species.

### Serial passage of RhCMV in HF results in improved viral replication

In order to assess whether RhCMV can adapt to improve its replication efficiency, we subjected RhCMV to 10 sequential passages in HF, initially in three replicates (Fig 2A). Analysis of the genomes of the passaged virus pools by Illumina sequencing revealed a mixture of wild type and mutant genomes. Prior to further analyses we performed plaque-purification of single isolates from each pool. We infected HF and HF$^{\Delta PKR}$ (MOI = 0.1) with EV1, a virus purified from pool 1, and with the wild type parent virus, and titered virus released into the medium over the course of 6 days (Fig 2B). EV1 produced greater than 10-fold more virus than did RhCMV on days 2, 4, and 6 post-infection of HF. Consistent with the data in Fig 1, the parent RhCMV replicated to higher titers in HF$^{\Delta PKR}$ compared to HF. We next analyzed the replication of two other viruses (EV2 and EV3) plaque-purified from the other two pools of virus that had been passaged 10 times in HF, in comparison to EV1 and the parent RhCMV. After infection of HF and HF$^{\Delta PKR}$ (MOI = 0.1), we titered virus released into the medium at 6 days. Like

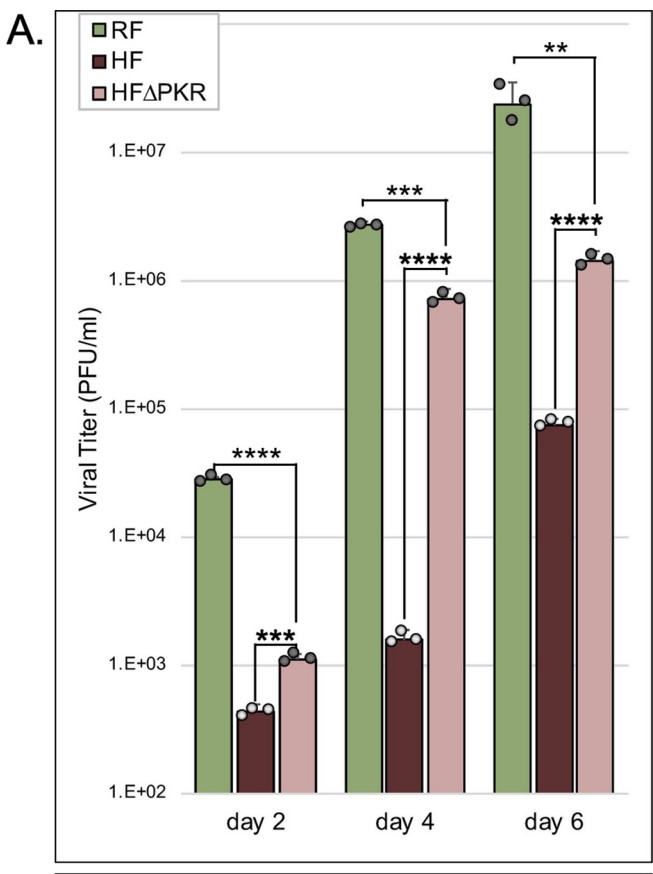

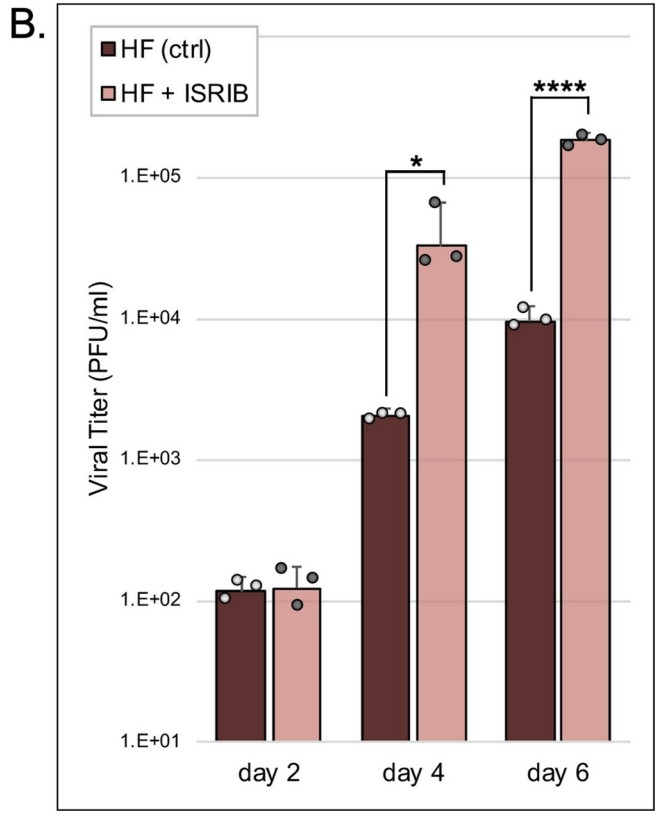

**Fig 1. Replication of RhCMV in HF is limited in part by human PKR.** (A) Measurement of RhCMV replication in RF, HF, and HF$^{\Delta PKR}$. RhCMV released into the medium over the course of six days following infection (MOI = 0.1) was titered on HF$^{\Delta PKR}$ cells (mean titer +/- standard deviation [SD]; n = 3; ○ = individual data points). Statistical significance for replication in HF$^{\Delta PKR}$ compared to that in RF and HF was assessed using an unpaired t test (**, P < 0.005; ***, P < 0.0005; ****, P < 0.0001). (B) Measurement of RhCMV replication in HF +/- 200 nM ISRIB. RhCMV released into the medium over the course of six days following infection of HF (MOI = 0.1) was titered on HF$^{\Delta PKR}$ cells (mean titer +/- standard deviation [SD]; n = 3; ○ = individual data points). Statistical significance for replication in the absence vs. the presence of ISRIB was determined using an unpaired t test (*, P < 0.05; ****, P < 0.0001).

EV1, both EV2 and EV3 produced significantly more progeny virus than the parental virus (Fig 2C) in HF but not in HF$^{\Delta PKR}$. These results reveal that the adaptation that emerged in all three viruses was specific for overcoming human PKR.

We noted that the evolved viruses tended to replicate less well in HF$^{\Delta PKR}$ than in HF. They also replicated less well than the parent RhCMV in HF$^{\Delta PKR}$. We detected a low but reproducible level of EV1 production by day 2 after infection of HF in this and other experiments (not shown), while no virus was produced at this time point after infection of HF$^{\Delta PKR}$. The titers of EV1 were significantly lower after infection of HF$^{\Delta PKR}$ compared to HF in the experiment shown in Fig 2B. In the experiment shown in Fig 2C, there was a similar trend toward lower replication of all three evolved viruses in HF$^{\Delta PKR}$ compared to HF, but the variations were not statistically different. These results suggest that while the evolved viruses clearly acquired changes that aided their replication in wild type cells, the adaptations may have compromised their fitness when PKR is absent.

## Gene amplification of the RhCMV PKR antagonist rTRS1 in the evolved viruses

To determine the genetic basis for the improved replication of EV1, EV2, and EV3 in HF, we analyzed the genomes of the parental RhCMV and the three plaque-purified evolved viruses by Illumina sequencing. The average read depth for each genome varied from ~100-800X. While we detected no single nucleotide polymorphisms present at a frequency greater than 1% (compared to the parent RhCMV sequence), we did find an ~11.8 kb deletion (from ~nt 182,846–194,560) and an approximate doubling of the read depth over the terminal 6.8 kb of the genome in all three evolved viruses (Fig 3A). This area of increased read depth encompasses nt 214,590–221,454 (plus the unannotated 764 bp terminal repeat) and includes the rTRS1 open reading frame (green arrowhead).

Mapping of reads onto the RhCMV 68–1 genome sequence (Genbank NC_006150) indicated that the amplified region was inverted and inserted into the region of the genome that had been deleted (Fig 3B). We used this analysis to design primers that span the predicted new left and right junctions between the inverted segment and the flanking wild-type regions (L1 & L2 and R1 & R2 in Fig 3B). Amplicons corresponding to these junctions were detected by PCR using each of the evolved viral DNAs as templates but were not detected with the parental RhCMV genome (Fig 3C). PCR with Rh44-specific primers served as a control for all viral genomes. Fig 4A shows a detailed view of the rearranged portion of the genome, including the positions of the right and left junction primers (used in Fig 3C) and encompassing the deleted and duplicated regions. The region of the genome deleted in the evolved viruses includes the C-terminal portion of the rh178.3 ORF (nt 182,846–182,966) through the C-terminal 2/3 of Rh195 (nt 194,048–194,560), while the duplicated genes include Rh 221 through rTRS1 (Rh230) and the 764 bp terminal repeat.

We purified the PCR products corresponding to the left and right junctional amplicons (Fig 3C) and analyzed them by Sanger sequencing. The sequences confirmed the results of

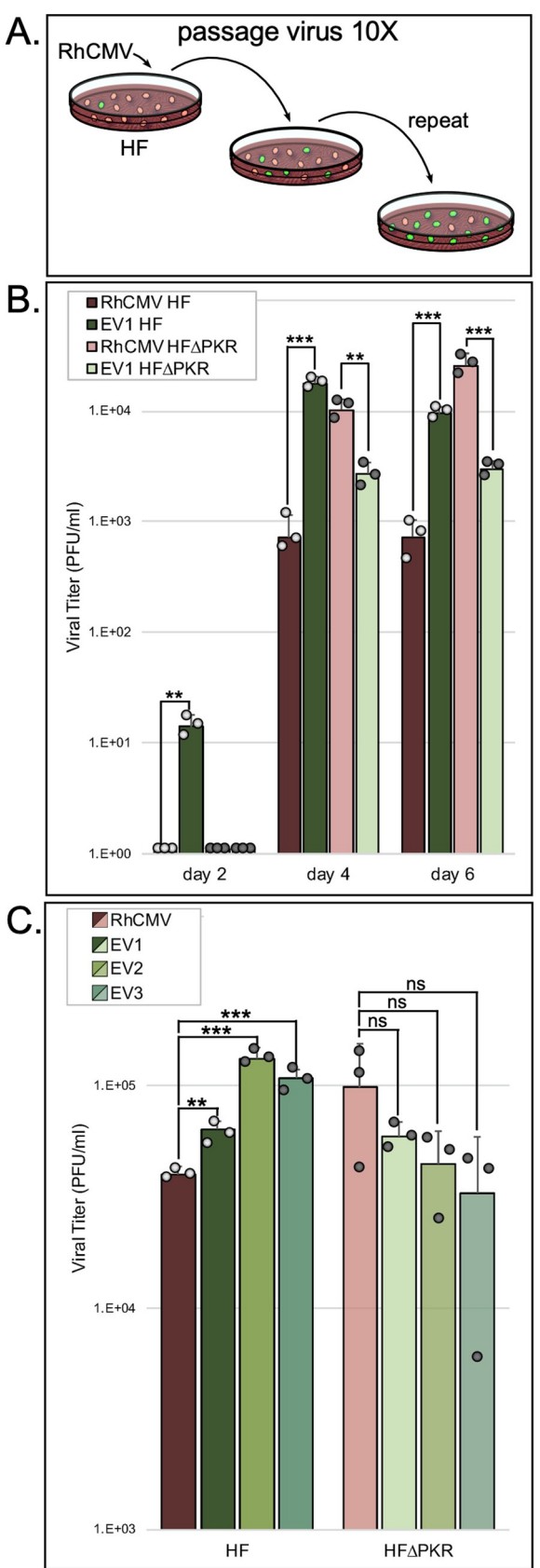

**Fig 2. Passaging of RhCMV in HF generates virus with improved replication efficiency.** (A) Schematic of the experimental evolution of RhCMV by serial passage in HF. (B) Measurement of parental RhCMV and EV1 replication in HF and HF$^{\Delta PKR}$. Virus released into the medium over the course of six days following infection (MOI = 0.1) was titered on HF$^{\Delta PKR}$ cells (mean titer +/- standard deviation [SD]; n = 3; ○ = individual data points). Statistical significance of differences between replication of RhCMV vs. EV1 in each cell type was determined using an unpaired t test (**, P < 0.005; ***, P < 0.0005). (C) Replication efficiency of three evolved viruses (EV1, EV2, EV3) compared to the parental RhCMV in HF and HF$^{\Delta PKR}$. Virus released into the medium on day 6 after infection (MOI = 0.1) was titered on HF$^{\Delta PKR}$ cells (mean titer +/- standard deviation [SD]; n = 3; ○ = individual data points). Statistical significance for replication of each evolved virus vs. parental RhCMV in each cell type was determined using an unpaired t test (**, P < 0.005; ***, P < 0.0005; NS, not significant).

Illumina sequencing, and are shown for the left (Fig 4B) and right (Fig 4C) junctions. Included in this figure are the results of sequencing for the initial virus pools (labeled "Pool"), indicating the most prevalent junction sequences present in the mixed populations. The left junctions varied slightly among the plaque-purified viruses, with up to 17 nucleotides deleted (in EV3). The right junctions for the vast majority of the mixed population reads and all three plaque-purified viruses are identical and create an in-frame fusion ORF between the 5´ 231 bp of the Rh195 ORF (in addition to 63 bp of linking sequence) and Rh221.

Interestingly, in the majority of the evolved pool sequences, the left junction of the inverted duplication is located ~34 bp downstream from what has been described as a *pac*1-like motif (Fig 4B) [26]. This sequence is very similar to the *pac*1 motif present near the genomic

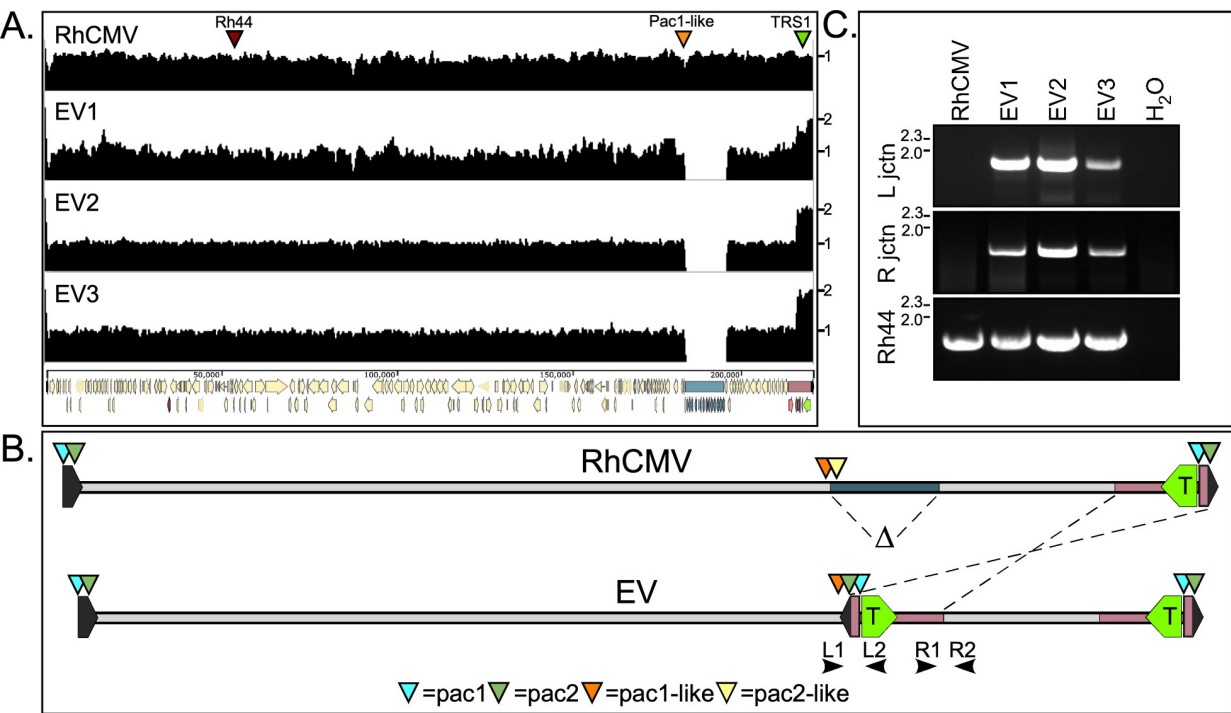

**Fig 3. Next-generation sequencing of the parental RhCMV and serially passaged viruses shows deletion and inverted duplication events in the three passaged lineages and suggests the functionality of a *pac*1-like motif.** (A) Sequence read depths for the RhCMV parent virus and three plaque-purified evolved viruses (EV1, EV2, and EV3). The genomic position of Rh44 (red arrowhead), the *pac*1-like motif (orange arrowhead) and rTRS1 (green arrowhead) are indicated. A gene map of the ~230 kb wild type RhCMV genome is shown at the bottom of panel (A). (B) Schematic showing the deletion and inverted duplication events that yielded the EV genomes. (C) PCR confirmation of the deletion/duplication junctions in the evolved virus lineages. Amplification using primers L1 and L2, and R1 and R2 (shown in panel B), yielded products spanning the newly formed left and right junctions, respectively, in the three plaque-purified evolved viruses. PCR of the Rh44 locus served as a control for the presence of viral DNA.

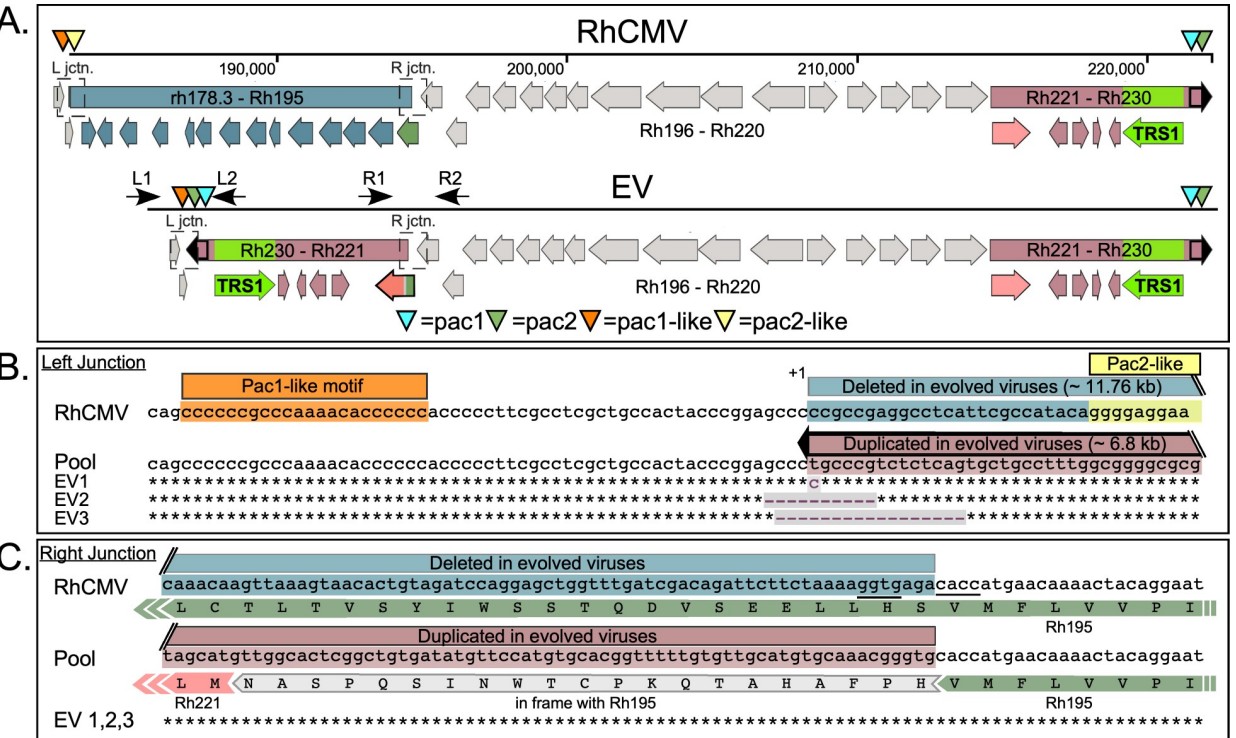

**Fig 4. Detailed analyses by Sanger sequencing of the new left and right junctions in EV1, EV2, and EV3.** (A) Map of the right portion of the parental RhCMV genome showing the regions and genes that are deleted and duplicated in the evolved viruses (EV). The positions of the primers used in (Fig 3B) are indicated by black arrows. The *pac1* (aqua arrowhead) and putative *pac2* (sage arrowhead) sequences within the genomic terminal repeat (black outlined) and the internal *pac1*-like (orange arrowhead) and *pac2*-like (yellow arrowhead) sequences are shown. Rh195 (dark green arrow), Rh221 (pink arrow), and the Rh195-Rh221 fusion protein formed at the right junction of the inverted fragment (green-gray-pink arrow) are also shown. (B and C) Sequence analysis of left and right deletion/duplication junctions. The junction-spanning PCR products from the left (B) and right (C) ends of the inverted duplication in EV1, EV2 and EV3 were purified and subjected to Sanger sequencing. The resulting sequences are mapped to the most common junction linkage obtained from Illumina sequencing of evolved virus pools. The protein sequence for Rh195 and the Rh195-Rh221 fusion protein are also indicated in (C).

terminus (Fig 5). *Pac1* is known to help direct the cleavage of replicating concatemers into unit length genomes by the viral terminase complex. Although a second component of the cleavage and packaging sequences, *pac2*, has not been clearly defined for RhCMV [26], there is a sequence near the end of the RhCMV genome that appears to be quite similar to *pac2* motifs described for other CMVs (Fig 5) [27,28]. Moreover, there is a *pac2*-like element, which is almost identical to the one located near the end of the genome, present ~60 nt downstream from the *pac1*-like motif in the parent genome. The left junction at which the duplicated region is inserted in evolved virus genomes is located precisely where the cleavage would be expected to occur relative to *pac1*-like and *pac2*-like, assuming these elements function like *pac1* and *pac2* at the ends of the genome. Though further studies are needed to clarify the role of the pac1- and pac2-like sequences, these observations suggest that cleavage, possibly by the terminase complex, may have facilitated the recombination events that resulted in the evolved virus genome structures.

## The inverted duplication present in serially passaged RhCMV generates a complex genome structure that enables genome isomerization

To clarify whether selection of these rearrangements was due to the effects of human PKR, we repeated our serial passaging experiment, this time including passages in HF$^{\Delta PKR}$ and RF as

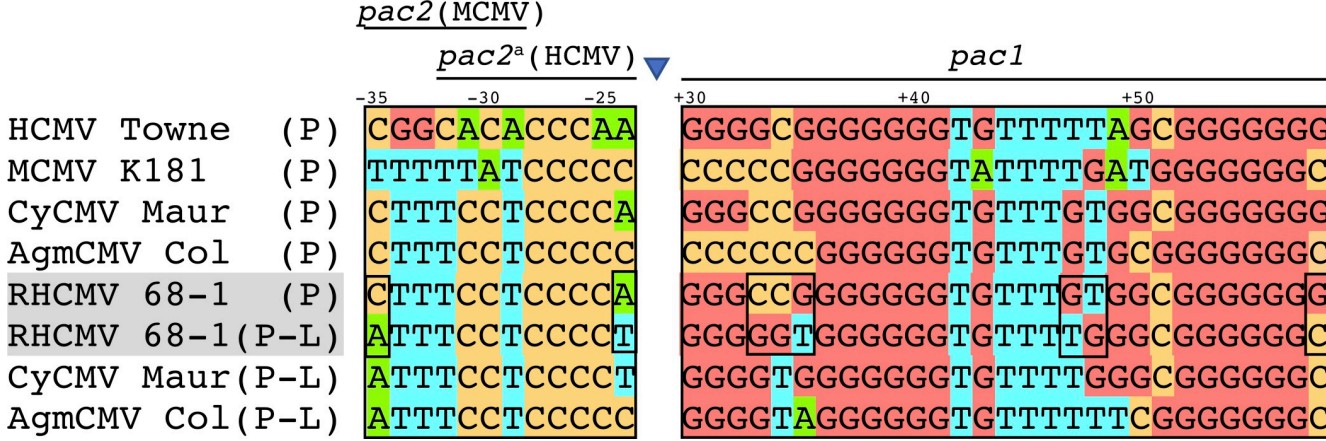

**Fig 5. Alignment of *pac*1, *pac*1-like, and proposed *pac*2 motifs in CMVs.** The *pac*1 and *pac*2 sequences (P) found at the ends of the genomes of HCMV (Towne), Murine CMV (MCMV K181) and three Old World monkey CMVs (rhesus, African green monkey and cynomolgus monkey) are aligned. These and other (not shown) Old World monkey CMVs each have similar *pac*1-like and *pac*2-like sequences (P-L). Because the *pac*1-like and *pac*2-like sequences are in an inverted orientation relative to the terminal *pac*2 and *pac*1 sequences, this alignment shows the *pac*1-like and *pac*2-like bottom strands, opposite to what is shown in Fig 4B. Nucleotides upstream of the predicted cleavage site are indicated with negative numbers, and those downstream are indicated with positive numbers, with cleavage occurring between nucleotides -1 and +1 (blue triangle).

well as in HF. Genomic DNA from the three new passage 10 virus pools from HF (pools 4, 5, and 6) were analyzed by Illumina sequencing, and all three pools demonstrated usage of the same left and right junction regions as were used for recombination in the first three viruses passaged in HF. No high frequency mutations were detected in the viruses passaged in HF$^{\Delta PKR}$ or RF, suggesting that these viruses are not subject to selective pressure in the absence of human PKR. We used Southern blot analyses to examine the structure of the genomes of all six virus pools passaged in HF, one pool each of viruses passaged in HF$^{\Delta PKR}$ and RF, as well as the genomes of plaque-purified EV1, EV2 and EV3 (Fig 6). Hybridization of *Hind*III-digested DNA from virus pools after passage in HF$^{\Delta PKR}$ and RF (Fig 6A, lanes 10 and 11) with a TRS1 probe showed a single band at 8.7 kb, consistent with the expected size of the fragment from the parental RhCMV genome (see also Fig 6B). While the rTRS1 duplication in the passaged viruses was expected to yield two specific bands (8.7 kb and 27.3 kb), analyses of the pooled and plaque-purified viruses passaged in HF revealed a surprisingly complex pattern of rTRS1-containing bands (Fig 6A, lanes 2–4, 6–8, 12–13 and 6B, EV1 lanes). Intriguingly, molecular weight analyses of the rTRS1-containing bands revealed that, like HCMV and other herpesviruses possessing complex E type genome structures, the genomes of the evolved viruses now appear to be capable of isomerizing such that the two repeat-flanked regions are able to invert with respect to each other to generate four combinations of genomic isomers (Figs 6C and S1). Note that the 14.9 kb and 11.5 kb fragments containing TRS1 are expected to be present in only ¼ of all genomes while the 8.7 kb and 21.3 kb fragments are present in ½ of the genomes, which can account for the differing intensities of these bands in the blots. Probing these same DNAs with the Rh44 gene yielded a single expected 15.1 kb fragment for each sample.

We also probed the RhCMV parent and the plaque-purified EV1 DNA with a fragment corresponding to sequences just upstream of the *pac*1-like motif that did not include any of the duplicated region (Fig 6B). This probe revealed a single band in the parent virus, demonstrating again that virus infecting RF does not utilize the *pac*1-like site for cleavage or isomerization at a detectable frequency. Conversely, in EV1, the *pac*1-like probe showed the bands expected to result from genome isomerization. As a result of the inverted duplication, the evolved

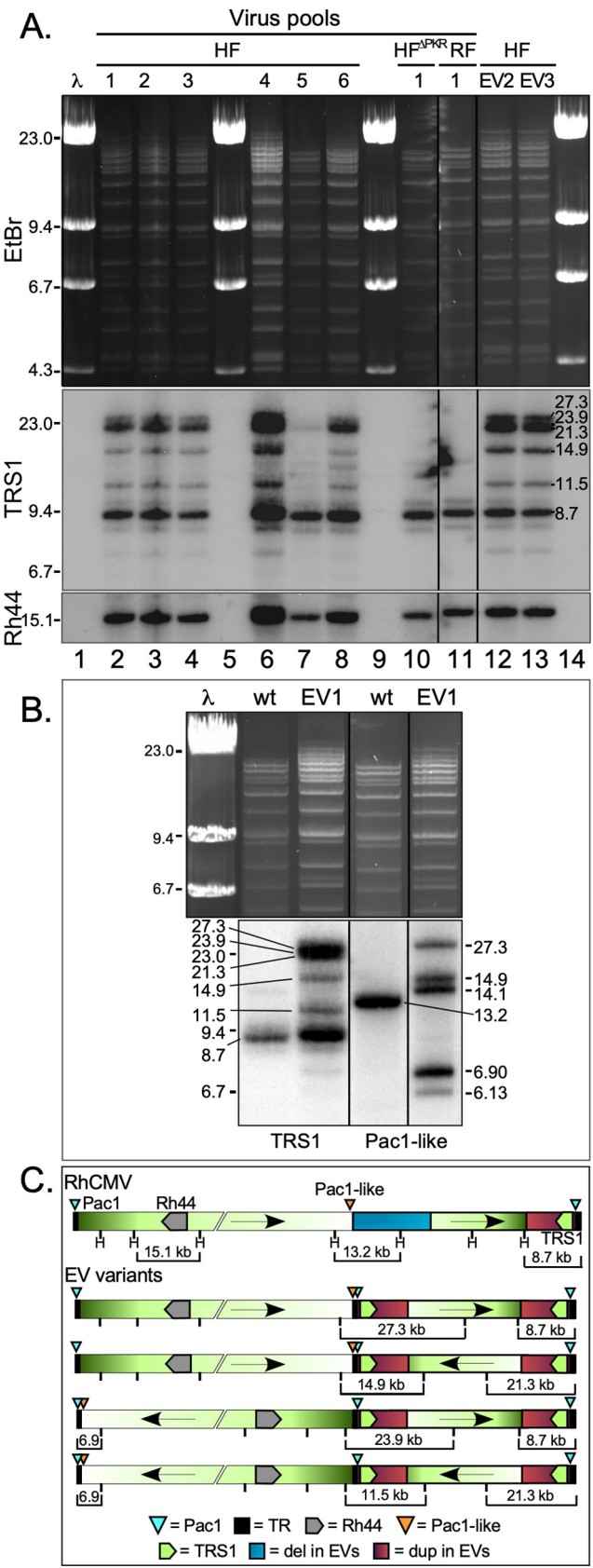

**Fig 6. Southern blot analyses of serially passaged RhCMV reveal that the inverted duplication generates a complex genome structure that enables genome isomerization.** (A) Analysis of viral genomic DNA from pools of viruses passaged in HF, HF$^{\Delta PKR}$, and RF, as well as plaque-purified EV2 and EV3 by Southern blotting. DNA purified from virus collected from the medium on infected cells was digested with *Hind*III then subjected to gel electrophoresis (EtBr staining, top panel) and Southern blotting with probes detecting rTRS1 (middle panel) and Rh44 (bottom panel). Molecular size markers are indicated on the left, molecular weights of specific hybridized bands are shown on the right. (B) Genomic analysis of RhCMV and plaque-purified EV1 by agarose gel electrophoresis and Southern blot. DNA was collected, digested, and electrophoresed as above (EtBr staining, top panel) and hybridized with probes for rTRS1 (bottom left) and a region just upstream of the *pac*1-like motif (bottom right). Molecular weights of markers and specific hybridized bands are shown. Blots are representative of at least three repeat experiments. (C) Schematic showing the inferred structures of parental RhCMV and the evolved virus genome isomers.

viruses contain an extra internal copy of the 764 bp terminal repeat sequence present at the ends of the wild type genomes, effectively creating an internal inverted repeat in the genome, similar to the '*a*' sequence found in herpesviruses that contain an E type genome structure. It is likely that this repeat, starting ~34 bp downstream of the *pac*1-like motif and containing the terminal *pac*1 sequence, enables isomerization of the large left genomic segment. A schematic of the parental RhCMV genome and the four genomic isomers generated by serial passage of RhCMV in human cells is shown in Fig 6C as well as in comparison to HCMV isomers in S1 Fig. A few bands that differ by ~800 bp (e.g. 14.1 and 14.9; and 6.13 and 6.9) might be the result of heterogeneity in the numbers of *a* sequence repeats in individual genomes, as has been observed in RhCMV and other herpesviruses [26,29,30]. Taken together these results suggest that the evolved viruses now effectively possess complex E type herpesvirus genome structure, complete with $U_L$ and $U_S$ segments.

## Gene duplication in the evolved RhCMVs results in rTRS1 overexpression

We next evaluated whether the gene duplication in the evolved viruses might lead to higher expression of rTRS1, which could account for more effective inhibition of PKR. We mock-infected or infected HF and HF$^{\Delta PKR}$ with RhCMV or plaque-purified EV1, EV2, or EV3 (MOI = ~3) for 72 h. We then collected lysates and performed immunoblot assays using antiserum directed against TRS1 as well as pRh44, which is expressed from a locus outside of the duplicated region. As shown in Fig 7, the evolved viruses expressed two to three times as much rTRS1 relative to pRh44 compared to the parental RhCMV virus in HF. Even in HF$^{\Delta PKR}$, where rTRS1 is not required to overcome PKR, the relative expression of rTRS1 compared to pRh44 was at least 2-fold higher following infection with the evolved viruses. Interestingly, in these and previous blots [21], rTRS1 appears as three distinct bands. While we have not yet investigated the basis for the differences in migration of these rTRS1 proteins, we suspect that they are due to posttranslational modifications. These results are consistent with prior studies using VACV in which gene amplification of a relatively weak PKR antagonist resulted in over-expression of the protein and improved viral replication [9,23]. In addition, we have previously shown that expression of extra rTRS1 from a second copy in the form of a cellular transgene is sufficient to rescue replication of recombinant VACV and HCMV containing a single rTRS1 gene [21,23].

## Increased levels of rTRS1 in the passaged viruses leads to reduced PKR pathway activation

To evaluate the effect of increased rTRS1 expression on human PKR activation, we mock-infected or infected HF with RhCMV or each of the three evolved viruses (MOI = 3), as well as RhCMV lacking rTRS1 (RhCMVΔrT) as a PKR pathway activation control. At 72 h post-infection we collected lysates and measured the accumulation of phosphorylated and total PKR and

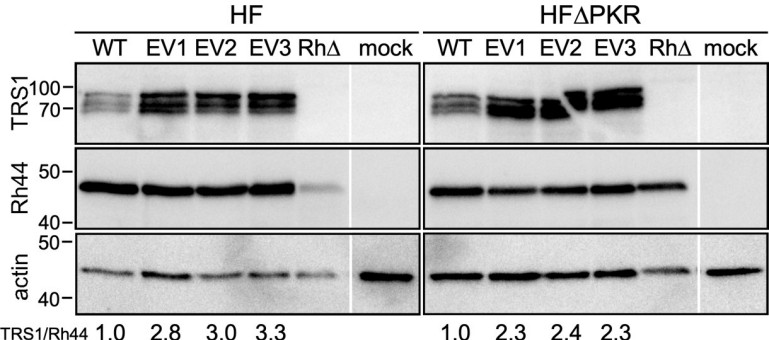

**Fig 7. The serially passaged evolved viruses produce more rTRS1 than the parental virus in HF and HF<sup>ΔPKR</sup>.**
Expression of rTRS1 from the parental RhCMV and three plaque-purified passaged viruses. HF and HF<sup>ΔPKR</sup> were
mock-infected or infected with parental RhCMV, EV1, EV2, EV3, or RhCMV[ΔrT] (MOI = 3), and at 72 h post-
infection the cells were lysed and equivalent amounts of protein (30 µg) were analyzed by immunoblotting with
antiserum directed against TRS1, Rh44, or actin as a loading control. The ratio of rTRS1 to Rh44 for each virus,
normalized to the ratio in cells infected with wild type RhCMV (WT), is indicated at the bottom. Blots are
representative of at least three repeat experiments.

eIF2α by immunoblot assays (Fig 8). The abundance of phospho-PKR and phospho-eIF2α rel-
ative to the total amounts of each protein was reduced in lysates of cells infected with EV1,
EV2, and EV3 compared to those cells infected with the parent RhCMV. Although the magni-
tude of the reduction in PKR and eIF2α phosphorylation by EV1, EV2, and EV3 varied in
repeats of this experiment, the results consistently showed that, compared to the parent virus,
all three evolved viruses blocked these measures of PKR pathway activation. As expected, PKR
and eIF2α were markedly phosphorylated in cells infected with RhCMVΔrT. Our previous
results, primarily using recombinant vaccinia viruses, demonstrated that unlike HCMV TRS1,
which blocks phosphorylation of both PKR and eIF2α, rTRS1 did not block phosphorylation
of human or African green monkey PKR but did inhibit phosphorylation of eIF2α [21,24]. In
contrast, these new results show that increased levels of rTRS1 during RhCMV infection lead
to lower levels of both phospho-PKR and phospho-eIF2α.

## Discussion

Viruses such as CMVs that have co-evolved with their primary host species for millions of
years generally have limited ability to replicate in different species because the viruses are not
optimally attuned to use dependency factors and to evade restriction factors in the divergent
host cells. Consistent with this perspective, HCMV is unable to replicate in rhesus cells and,
although RhCMV can replicate in HF, it does so much less efficiently than in RF. Our finding
that RhCMV replicates significantly better in HF<sup>ΔPKR</sup> than in HF and in cells treated with
ISRIB compared to untreated cells (Fig 1) demonstrates that a substantial limitation to
RhCMV replication in HF results from its inability to completely block the PKR pathway in
human cells.

Serial passage of RhCMV through HF resulted in viruses with improved replication effi-
ciency in HF. The loss of any replication benefit in HF<sup>ΔPKR</sup> indicated that these evolved viruses
had adapted specifically to eliminate the repressive effects of human PKR. In fact, we observed
that the evolved viruses generally replicated to a higher level in HF than in HF<sup>ΔPKR</sup>. This obser-
vation suggests that PKR may serve some pro-viral function, provided that the virus is able to
block its major inhibitory impact on translation. However, this effect was not consistently sig-
nificant (Fig 2C) and in prior studies of HCMV we did not detect any significant differences
between replication in wild type HF vs HF<sup>ΔPKR</sup> [20]. Regardless, our finding that the passaged

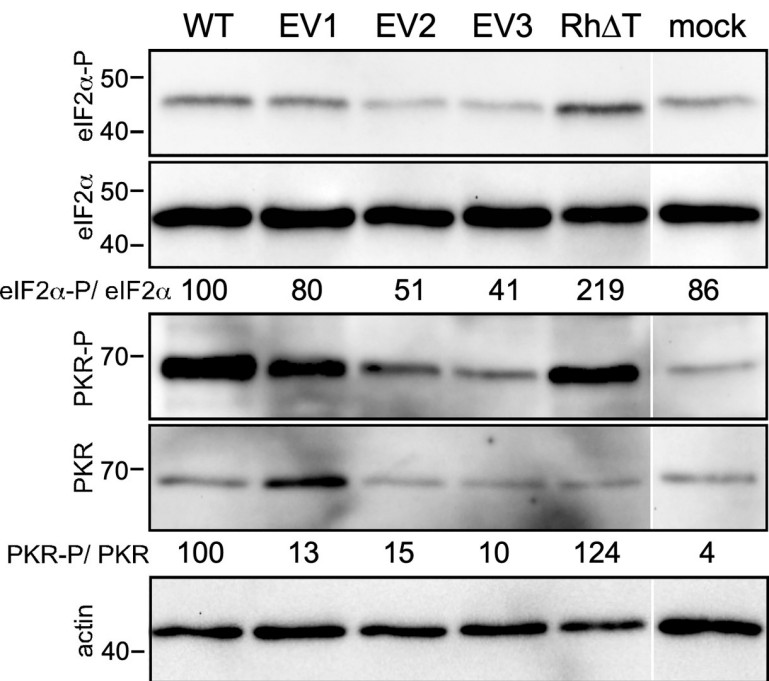

**Fig 8. Effects of rTRS1 expression on PKR and eIF2α phosphorylation.** Pathway analysis in HFs infected with the parental RhCMV or plaque-purified passaged viruses. HF were mock infected or infected with RhCMV, EV1, EV2, EV3, or RhCMV[ΔrT]. At 72 h post-infection the cells were lysed, and equivalent amounts of protein (30 μg) were analyzed by immunoblotting with antibodies directed against total and phospho-PKR, total and phospho-eIF2α, and actin as a loading control. The levels of eIF2α and PKR phosphorylation relative to total eIF2α and PKR in each sample, normalized to the ratios in wild type RhCMV (WT) infected cells, are indicated below the corresponding immunoblots. Blots are representative of at least three repeat experiments.

viruses show improved ability to counteract human PKR supports the conclusion that PKR is one key restriction factor underlying the limited replication of RhCMV in HF.

Our analyses of the genomes of the evolved viruses revealed a reproducible duplication of a segment of the genome containing the gene encoding the RhCMV PKR antagonist, rTRS1. Our Illumina sequencing data are consistent with the average genome containing two copies of rTRS1, but there may be some heterogeneity in the copy number among individual genomes. In accordance with a gene dosage model, these viruses expressed 2-3-fold higher levels of rTRS1 than the parent virus and they were more effective in preventing PKR pathway activation (Figs 7 and 8). These findings are consistent with prior studies using VACV in which genes encoding weak PKR antagonists were amplified by serial passage [9,23,31,32]. In a VACV lacking both of its natural PKR antagonists, E3L and K3L, rTRS1 was unable to support any VACV replication in human cells. Passage through an African green monkey cell line in which a single copy of rTRS1 allowed a low level of replication resulted in amplification of the rTRS1 gene. The VACVs with amplified rTRS1 were then able to replicate in human cells. This model showed that passage through a semi-permissive intermediate host may facilitate a cross-species transmission event, as has been postulated to be the pathway leading to at least several zoonotic epidemics [33,34].

Among the differences between prior studies of rTRS1 in the context of recombinant VACV and the RhCMV studies reported here is the mechanism by which rTRS1 blocks PKR. In the VACV system, expression of rTRS1 did not prevent PKR autophosphorylation in permissive African green monkey (BSC40) cells, but it did prevent eIF2α phosphorylation, suggesting that binding of rTRS1 to the African green monkey PKR variant does not block PKR

autophosphorylation but does block its eIF2α kinase activity [23,24]. However, in the context of RhCMV, overexpression of rTRS1 caused a marked decrease in PKR autophosphorylation as well as eIF2α phosphorylation (Fig 7), similar to the effects of the HCMV antagonists, TRS1 and IRS1 [11,20,35,36]. The basis for these differing effects of rTRS1 between the VACV and CMV systems is unknown, but the vast differences in replication kinetics and impacts on host mRNA translation between VACV and CMVs might underlie these observations. Regardless, these new data suggest that the mechanism by which the PKR antagonists encoded by HCMV and RhCMV bind to PKR, as well as their function in blocking PKR autophosphorylation, is likely quite similar.

Another difference between the results of experimental evolution in the VACV vs RhCMV systems is the characteristics of the gene amplification events. In two independent VACV studies, viruses that evolved to overcome PKR had a heterogenous number of tandem head-to-tail copies of the PKR antagonist gene [9,23]. Similar tandem amplification events were detected in VACV subjected to other selective pressures [37,38]. In this new study, we detected only a single duplication event, and it occurred at a distal site and in an inverted orientation. The number of copies may differ in part because VACV has a less structured capsid that can likely tolerate larger insertions compared to CMVs [39]. In fact, some genomes emerging in the VACV experiments increased in size by as much as 10% [9]. The evolved RhCMVs have a large duplication (6.8 kb) but an even larger deletion (11.8 kb), suggesting that the deletion may have been required to allow genome packaging. Clearly, the deleted genes, Rh196-220, are not essential for viral replication in cell culture. Indeed, several of these genes are homologous to HCMV genes needed for evading cellular immune responses, which would not be expected to be required for replication in cell culture [2,40,41]. However, the evolved viruses tended to replicate less well than the parent virus in HF$^{\Delta PKR}$ (Fig 2), suggesting that adaptation to PKR did incur a minor fitness cost. It is possible that one of the deleted genes is necessary for efficient replication in cell culture. Alternatively, the duplication of a gene such as Rh223, the homolog of HCMV US30, which has been reported to be suppressive for HCMV replication in HF [40,42], might have a negative effect on replication under these conditions.

The likelihood that packaging of the RhCMV genome is constrained by capsid size limitations raises the question as to whether a gene duplication event such as we observed in cell culture would occur in nature. When the HCMV genome was first sequenced, it became clear that the virus has multiple gene families, which presumably arose through gene amplification events [43,44]. The hominoid and New World monkey CMVs encode TRS1 and IRS1, each belonging to the US22 gene family, which contains ~12 members. The identity of the N-terminal ~550 amino acids of HCMV TRS1 and IRS1 suggests that they arose by a gene duplication event during evolution, similar to what we observed in our experimental RhCMV system. Despite some divergence at their C-termini (which retain ~40% amino acid identity), HCMV TRS1 and IRS1 appear to have largely redundant essential functions in inhibiting PKR [11,35,45]. In other cases, the individual gene copies have acquired new or subspecialized functions. For example, two adjacent US22 family genes encoding murine CMV genes m142 and m143 act in a complex to block murine PKR [14,15,46–48]. Rat CMV encodes close homologs of the mouse CMV genes [49], leading us to hypothesize that m142 and m143 arose by tandem duplication and subsequent sub-functionalization during rodent CMV evolution. Thus, despite the potential of there being genome size constraints, gene duplication events have very likely occurred repeatedly during CMV evolution.

We were surprised at how the genomic structures that emerged following serial passage of RhCMV in HF were nearly identical in multiple replicates. Intriguingly, one end of the inverted duplication occurred ~34 bp downstream from a site known as *pac*1-like. This element was noted in a prior analysis of the RhCMV genome to closely resemble the *pac*1

cleavage and packaging signal located near the end of the genome [26]. A second motif known to be important in cleavage and packaging, *pac*2, is positioned about 60 nt from *pac*1 prior to genome maturation. Cleavage occurs between the motifs so that *pac 1* and *pac*2 end up near opposite ends of the mature genome. Interestingly, we noticed a second copy of a presumed *pac*2 motif (pac2-like) ~60 bp downstream from the *pac*1-like sequence in the wild type genome (Figs 4B and 5). These observations suggest that the left end of the rearranged genome may have been enabled by occasional cleavage of the RhCMV genome between the *pac*1-like and *pac*2-like elements, presumably by the same viral terminase complex responsible for cleavage at the normal genome end. At present, we do not have a clear hypothesis to explain the formation of the new right junction in these viruses. We do note the presence of a short palindrome (5'- GGTG-N3-CACC -3') in the parent genome (Fig 4C) and following the rearrangement, a similar palindrome 5'- GGTG-CACC -3' is created at the right junction, suggesting the possibility that DNA structure and/or binding proteins may have facilitated the use of this site.

The fact that we selected viruses with almost the same structure in six separate lineages suggests that the recombination event that generates this structure might occur relatively frequently. An alternative explanation is that the original recombinant arose by a rare event but happened to be present in the starting stock of virus. Illumina sequencing of the parent RhCMV genome (with a read depth of ~1500) detected one read that appears to arise from an inverted duplication event, suggesting that it may have been present in the starting pool. On the other hand, the left end junctions in plaque-purified EV1, EV2 and EV3 are not identical (Fig 4B), and a variety of slightly different left junction linkages are present in the pre-purification pools of these viruses. Also, the second set of passaged viruses, pools 4, 5 and 6, all showed predominant left junction connections that differ from those found in EV1, EV2 and EV3. Finally, in our Illunima sequencing data, we detected the rearranged left junction in DNA from an entirely independent stock of wild-type RhCMV (non BAC-derived RhCMV), indicating that the recombination event leading to the evolved virus structures likely was not unique to the one particular stock used in these experiments. Thus, there appears to be a propensity for recombination to occur downstream from *pac*1-like, enabling rTRS1 duplication to occur at a low background frequency. Under the appropriate selective pressure, in this case provided by human PKR, the recombinant virus will out-compete the wild type virus upon serial passage.

In support of the functionality of the *pac*1-like motif, recent reports have documented the presence of highly conserved G-quadruplexes, which are non-canonical secondary structures formed by stacked guanine tetrads, in the *pac1* signals of all human herpesviruses [50]. These structures require four islands of two or more guanines, separated by single-stranded loops of usually one to seven nucleotides. With the homology between the *pac1* and *pac1*-like sequences, it seems likely that *pac1*-like could also form G-quadruplexes. A recent report highlighted the enrichment of G-quadruplexes in the regions flanking DNA breakpoints in HSV-1 and demonstrated that they are sites of recombination [51]. Additionally, reports that HSV-1 recombination factors such as ICP8 have been shown to co-localize with G-quadruplexes, and that UL12, an alkaline nuclease that binds the cellular double strand break-sensing MRN complex, co-localizes with ICP8 [52] suggest that G-quadruplexes may play a role in recruiting recombination machinery during herpesvirus DNA replication.

One intriguing additional consequence of the inverted duplication is the creation of RhCMV with a herpesvirus type E genome, similar to the structure of hominoid and New World monkey CMVs and even the more distantly related herpes simplex viruses (Figs 6 and S1) [44,53,54]. In these other cases, the repeat elements contain genes encoding PKR antagonists. This observation raises the possibility that adaptation to PKR, as occurred in our

RhCMV cell culture experiments may have contributed to the evolution of the herpesvirus type E genome.

In summary, RhCMV appears to be poised to adapt to challenges by undergoing gene duplication, taking advantage of a cryptic cleavage-packaging signal embedded in the genome. All Old World monkey CMVs analyzed so far share these *pac*1-like sequences but are missing the internal repeat *a* sequences found in hominoid and New World monkey CMVs. Although we do not know if the internal repeats arose independently in hominoid and New World monkey CMVs, or were lost in the Old World monkey lineage, our results demonstrate the relative ease with which RhCMV can adapt to a host restriction factor in cell culture. Based on the presence of gene families in CMV, we suspect that a similar genomic rearrangement might have occurred and been selected for repeatedly during CMV evolution.

## Materials and methods

### Cells

All cells were maintained in Dulbecco's modified Eagle's medium supplemented with 10% NuSerum (BD Biosciences). HF were obtained from Denise Galloway (Fred Hutchinson Cancer Research Center), and telomerase-immortalized RF were obtained from Peter Barry (University of California, Davis). HF$^{\Delta PKR}$ were described previously [21].

### Viruses, BAC recombinant viruses, and infections

RhCMV (strain 68–1, ATCC VR-677) and RhCMV BAC DNA derived from the strain 68–1 virus were obtained from Peter Barry (University of California, Davis). The BAC-derived RhCMV was reconstituted by transfection into RF and both viruses were propagated on RF. The RhCMV[ΔrT] mutant was generated by deleting Rh230 (rTRS1) from the RhCMV 68–1 BAC as described previously [55]. The virus was reconstituted by transfecting purified BAC DNA into RF expressing rTRS1 and was later plaque purified three times on HF$^{\Delta PKR}$. For each growth curve, cells were infected, and at 1 h post-infection, the cells were washed 3 times with PBS, after which the medium was replaced. For measurement of RhCMV replication +/- ISRIB, 200 nM ISRIB (in 0.2% DMSO) was added at 1 h post-infection. Growth curves were performed by collecting the supernatants from cells infected with the indicated viruses at the specified times, and titers were determined on HF$^{\Delta PKR}$. Data for all viral growth curves are representative of at least three separate experiments.

### Experimental Evolution of RhCMV

For each passage, 100 mm dishes were seeded with wt HF (~5 x 10$^6$ cells/dish). For each sample, triplicate dishes of cells were infected with virus at a multiplicity of infection of 0.1 pfu/cell. When the cells were nearly 100% infected, the cells were pelleted, resuspended in 1 ml of DMEM + 10% Nu serum, and virus was released by three freeze/thaw cycles. Viral titers were calculated by performing plaque assays in HF$^{\Delta PKR}$ cells between each passage. After ten passages, larger viral stocks were made, in some cases following three rounds of plaque purification. Genomic DNA was harvested from the infected cell medium and prepared for sequencing by proteinase K digestion, phenol:chloroform extraction, and two rounds of precipitation.

### Genomic analyses

Libraries were prepared using the KAPA HyperPlus kit with 100ng of genomic DNA as input and sequenced on an Illumina MiSeq. Reads were trimmed using Trimmomatic v0.39 and mapped to the Macacine herpesvirus 3 (rhesus cytomegalovirus strain 68–1) reference genome

NC_006150 using the Geneious read mapper (PMIDs 24695404, 22543367). Sequencing reads were deposited in NCBI BioProject PRJNA660187. RhCMV NC_006150 and the following sequences obtained from Genbank were used for comparison of the pac and pac-like regions (see Fig 5): Cynomolgus macaque cytomegalovirus strain Mauritius (KP796148), Cercopithecine betaherpesvirus 5 strain Colburn (FJ483969), Human herpesvirus 5 strain Merlin (NC_006273.2), Murid betaherpesvirus 1 (NC_004065.1).

PCR reactions were carried out using Phusion polymerase (NEB) according to the manufacturer's recommendations with the following primers: left junction, #2507 (5'- GG CAGA GAAGGACGAGATTAAG -3') and #2508 (5'- GAACGCCGAAGCAGTAGAA -3'); right junction, #2492 (5'- GGTGATGCAGGTGTATGGTT -3') and #2514 (5'- ATTCCGG CTGCC ATACTTATC -3'); Rh44 #2476 (5'- gtcggatccgtccggcggccATGGAGAGGA AAGCGCGTTTTA CCCG -3') and #2477 (5'- ggccgccactgtgctggatatctgcagaattgcccct TGTACATTTCTGCTTTTT GCTGC -3'). PCR products were purified by gel isolation or using a QIAquick PCR purification kit (Qiagen) prior to Sanger sequencing.

For Southern analyses, viral DNAs were purified from cell-free virus from medium collected off of infected cells, digested with HindIII, following which the fragments were resolved by agarose gel electrophoresis and transferred to supported nitrocellulose. Hybridization probes consisted of [$^{32}$P] labeled random-primed PCR products generated using the following primers: rTRS1, #723 (5'- CCAAAGATCTACCATGCGTCCTCACCGCTCGCCA -3') and #724 (5'- GCACGGGACGATGAGAACACCAT -3'); upstream of pac1-like #2539 (5'- GGCG CGAAACACGCGTTTG -3') and #2540 (5'- CTGAAAATGGCAAGTGGCCG -3'); Rh44, #2476 and #2477 (see above).

## Protein immunoblot analyses

Samples for immunoblotting were prepared by lysing cells with 2% SDS. The lysates were sonicated in a bath sonicator to disrupt nuclear DNA, and then proteins were separated by SDS-polyacrylamide gel electrophoresis on gels containing 0.5% 2,2,2-trichloroethanol to allow stain-free fluorescent visualization of proteins (50), transferred to a polyvinylidene difluoride (PVDF) membrane (Millipore), and probed with the indicated antibodies using the Western Star chemiluminescent detection system (Applied Biosystems) according to the manufacturer's recommendations. The antibodies used in these experiments included eIF2α L57A5 (number 2103), phospho-eIF2α Ser51 (number 3597), and PKR D7F7 (number 12297), all from Cell Signaling Technology, as well as phospho-PKR T446 (ab32026; AbCam), actin (A2066; Sigma), and RhCMV Rh44 (52). Polyclonal rabbit antiserum that recognizes the RhCMV TRS1 dsRNA-binding domain (α999), has been described previously (8). All the purchased antibodies were used according to the manufacturers' recommendations. Immunoblot images were captured and quantified with a ChemiDoc Touch imaging system and Image Lab software (Bio-Rad Laboratories, Hercules, CA).

## Statistical analyses

All statistical analyses were performed using an unpaired, two-tailed t test. If unequal variances were observed for unpaired sample sets (F test for unequal variance), an unpaired t test with Welch's correction was performed. Statistical analyses were performed using Prism 7 software (GraphPad).

## Supporting information

**S1 Fig. Comparison of the genomic organization and isomers of HCMV, wild type RhCMV and evolved RhCMVs.** The inverted duplication of the end of the RhCMV genome,

including the terminal repeat, that generated the evolved viruses reported in this manuscript, results in a type E genomic organization that is very similar to that found in HCMV as well other hominoid and New World Monkey CMVs. The parental RhCMV, like other Old World monkey CMVs, has a type A genomic structure with direct terminal repeats but no internal repeat sequences.

(TIF)

## Acknowledgments

We thank Hong Xie (University of Washington) for sequencing library preparation, Peter Barry (University of California, Davis) and Denise Galloway (Fred Hutchinson Cancer Research Center) for reagents and Harmit Malik (Fred Hutchinson Cancer Research Center) for helpful discussions.

## Author Contributions

**Conceptualization:** Stephanie J. Child, Adam P. Geballe.

**Data curation:** Stephanie J. Child, Alexander L. Greninger, Adam P. Geballe.

**Formal analysis:** Stephanie J. Child, Alexander L. Greninger, Adam P. Geballe.

**Funding acquisition:** Adam P. Geballe.

**Investigation:** Stephanie J. Child, Alexander L. Greninger, Adam P. Geballe.

**Methodology:** Stephanie J. Child, Alexander L. Greninger, Adam P. Geballe.

**Validation:** Stephanie J. Child.

**Writing – original draft:** Stephanie J. Child, Adam P. Geballe.

**Writing – review & editing:** Stephanie J. Child, Alexander L. Greninger, Adam P. Geballe.

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
