## [Decision Letter · Decision Letter 0]

5 Dec 2020

Dear Dr. Geballe,

Thank you very much for submitting your manuscript "Rapid adaptation to human protein kinase R by a unique genomic rearrangement in rhesus cytomegalovirus." for consideration at PLOS Pathogens. As with all papers reviewed by the journal, your manuscript was reviewed by members of the editorial board and by several independent reviewers. The reviewers appreciated the attention to an important topic. Based on the reviews, we are likely to accept this manuscript for publication, providing that you modify the manuscript according to the review recommendations.

All three reviewers spoke highly of your results. They ask for minimal changes to the text. While one reviewer asks for additional experiments, I'm not convinced they are necessary. In any rebuttal letter, please make sure to address all reviewer comments.

Sincerely,

Robert F. Kalejta

Associate Editor

PLOS Pathogens

Blossom Damania

Section Editor

PLOS Pathogens

Kasturi Haldar

Editor-in-Chief

PLOS Pathogens

orcid.org/0000-0001-5065-158X

Michael Malim

Editor-in-Chief

PLOS Pathogens

orcid.org/0000-0002-7699-2064

All three reviewers spoke highly of your results. They ask for minimal changes to the text. While one reviewer asks for additional experiments, I'm not convinced they are necessary. In any rebuttal letter, please make sure to address all reviewer comments.

Reviewer Comments (if any, and for reference):

Reviewer's Responses to Questions

**Part I - Summary**

Reviewer #1: The manuscript by Child et al. characterizes the selective pressures that enable variants of rhesus cytomegalovirus (RhCMV) to replicate with greater efficiency in cultured cells derived from a non-rhesus host (i.e., human fibroblasts). The basis for this study is that CMV representatives exhibit a strict species specificity for productive infection. Human CMV will only productively infect human cells, rodent CMVs efficiently infect the appropriate species of cells. RhCMV efficiently infects rhesus cells, but will also infect some human cells, albeit with reduced efficiency compared to infection of rhesus cells. Barriers to efficient replication cover multiple restriction points beginning attachment and entry, to transport of the viral genome to the nucleus, to temporal gene transcription and translation, and to viral assembly and release. Herpeseviruses, in general, and CMVs, in particular, have co-evolved with their particular host; viruses and hosts have each shaped the other’s genome in an “arms race” that is ongoing since the progenitor herpesvirus arose ~400 MYR ago, and which enables a mostly stable virus-host detente for the lifetime of the infected host.

This group has previously demonstrated that RhCMV is able to cross species-specific barriers to replicate in human cells through a RhCMV-mediated attenuation of the cellular protein kinase R (PKR), a key antiviral factor, by the RhCMV-encoded rTRA1 protein, the ortholog of human CMV TRS1. This current manuscript greatly leverages their previous work by investigating the outcomes of serially passaging RhCMV in human cells to determine how RhCMV adapts in the context of repeated confrontations with human restriction factors, such as PKR. The experimental design is fundamentally sound and easily fits into a rubric of “simple but elegant”. The authors convincingly confirm their prior studies that human PKR is a restriction factor for RhCMV replication in human fibroblasts (HF) by demonstrating that infection of RhCMV in HFs is increased through either targeted genetic deletion of the PKR locus or drug-mediated inhibition of PKR function. The data are absolutely convincing and fully support the author’s conclusion that, “human PKR restricts RhCMV replication”.

To investigate how RhCMV adapts to repeated selective pressures imposed by restriction factors, such as PKR, parallel cultures of RhCMV-infected HFs were serially passaged 10 times in HFs without plaque purification. What is especially attractive about the authors’ approach is that it should enable the emergence of any RhCMV variants with enhanced growth advantage above the input parental strain. What is especially remarkable is that this approach selected in six replicate examples the identical, or nearly identical genetic variants. In particular, the 10-fold repeated passaging selected for variants that (a) grew with enhanced fitness in HFs, compared to the input parental virus, and (b) contained the same genetic rearrangement in which a portion of the RhCMV genome was deleted and replaced by a duplication of the terminal region of the linear RhCMV genome encoding the rTRS1 gene. The different variants differed only by the precise location of the left end of the deleted region. Surprisingly, no other genetic variants >1% were detected by Illumina sequencing (100-800-fold read depths). The data from these experiments are solid, and there is no doubt about the validity of the results.

What the authors have seemingly uncovered is that the primary restriction point for cross-species infection of cultured HFs by RhCMV is PKR since the relatively identical genetic rearrangements were repeatedly observed. Since there are multiple potential restriction points to efficient infection of HFs by RhCMV, the results imply that PKR is at the apex of a putative hierarchical structure of species restriction. In another truly surprising finding is that the particular genetic rearrangement led to a novel finding for RhCMV genome replication. In particular, the rearrangement resulted in a complex E type genome structure, like that of human CMV, capable of undergoing genome isomerization. The parental RhCMV genome does not exhibit the E type genome structure and does not isomerize. What is so appealing about this manuscript is that the results open many new avenues if investigation that couldn’t have been envisioned a priori. One future experiment that could be conducted, and is way beyond the scope of this work, is whether serial passage in rhesus fibroblasts would back select for a non-E type genome structure (given the fact that ~11 kbp were deleted during serial passage in HFs).

The authors further demonstrate that the genomic rearrangement duplicates the rTRS1 gene, resulting in increased expression of rTRS1 in infected cells and consequent reduction of PKR functional activity. This result again demonstrates the functional linkage in rTRS1 targeting PKR activity as a primary restriction point to RhCMV infection.

In sum, this is an excellent manuscript and clearly worthy for publication in PLOS Pathogens. There are no deficiencies in the experimental design, results, and date interpretation. Given the current globally shared experience of SARS CoV-2, these results shed light into the mechanisms pathogens employ to overcoming species restrictions. While the results may not be directly relevant to the initial zoonotic spread of SARS CoV-2 from an animal species to humans, the results provide guidance into the restriction factors that pathogens may target.

Reviewer #2: The authors study how rhesus CMV might evolve cross-species transmission with a focus on overcoming inhibition of virus replication by antiviral human protein kinase R (PKR). Using protocols of experimental evolution in human fibroblast cells they find that RhCMV undergoes recombination events leading to an inverted duplication of the TRS1 gene, resulting in greater virus replication and more TRS1 protein and less phosphorylation of PKR and its substrate eIF2alpha. They link the recombination events to nearby, repetitive Pac-like sequences and find that the adapted isolates they characterized behave like E class CMVs as judged by 4 isomer genome populations following the duplication events. The manuscript is clearly written and describes a nice set of experiments and follow-up analysis. While virus adaptation through a duplication event is an interesting result, the class E like behavior of the adapted isolates reminiscent of hCMV is surprising and presents an intriguing scenario for the evolution of more complicated E class herpesviruses like hCMV, a result of broad interest in virus evolution.

Reviewer #3: Child and colleagues describe a study in which they investigate a role of the IFN-inducible restriction factor PKR in the species specificity of cytomegaloviruses (CMV), a topic to which this research group has made many key contributions. Rhesus macaque CMV (RhCMV) replicates on human fibrobalsts (HF) but to much lower levels than on Rhesus fibroblasts (RF). Interestingly, deletion of PKR from HF rescues viral growth as does treatment with a factor that impairs PKR function, although not to levels observed in RF. In light of this, the authors utilize experimental in vitro evolution of RhCMV by serial passage on HF and find that 1) adaptation occurs that allows improved replication on these cells; and 2) viral genomic changes associated with this involve duplication of a region encoding TRS1, a protein necessary and sufficient for evading PKR inhibitory processes. Intriguingly, 3 replicate evolution experiments led to similar genomic changes yet parallel evolution experiments on RF or HF lacking PKR do not acquire these. This suggests a TRS1 gene dosage mechanism may be mechanistically linked with the ability of the derivative to grow better on HF. Consistent with this, increased levels of TRS1 protein are observed by immunoblot in the evolved relative to parental virus. In addition, activation of PKR as indicated by phosphorylation of the protein appear diminished when HF are infected with evolved versus parental virus or TRS1-deficient RhCMV. Overall this represents an interesting study with implications for our understanding of the processes and evolutionary events involved in host adaptation by herpeseviruses. It also should be of general interest given the potentially devastating impact of spontaneous coronavirus host range expansion as generalizable principles may be evident across virus types. A few issues should be addressed prior to publication however.

**Part II – Major Issues: Key Experiments Required for Acceptance**

Reviewer #1: No additional experiments required for acceptance.

Reviewer #2: To better highlight the proposed Pac-based duplication and E class isomerizations of genomes, a diagram illustrating hCMV isomers might be added to make a more direct comparison to RhCMV. This is partially illustrated in Figure 6, but missing the hCMV comparison, which might be developed further. For example, is TRS1/IRS1 in hCMV a likely outcome of a similar duplication? Might it be possible to get hCMV to “revert” to a more simple single isomer based on the RhCMV result in the other direction? Is there any phylogenetic evidence that the Pac sequence arrangements found in RhCMV are ancestral?

In some of the follow-up experiments the authors describe experimental evolution of RhCMV in rhesus fibroblasts and human fibroblasts lacking PKR. Did the viruses improve in ability to replicate after these protocols?

Is EV3 better at blocking phosphorylation of PKR and eIF2alpha compared to EV1 and EV2? Are the blots (westerns and southerns) representative of multiple experiments?

Reviewer #3: 1. The data are consistent with TRS1 gene dosage via coding region duplication being the primary selected phenotype that allows enhanced growth on HF. However, given that other genomic changes occurred and their effects may also make contributions as well as the fact that growth is not totally restored on PKR-deficient cells, additional measures should be pursued to establish the specific necessity of TRS1-PKR relationship. For instance, demonstrating that ectopic overexpression of TRS1 rescues growth of parental RhCMV on HF would enhance the strength of this conclusion. Alternatively, perhaps it’s possible to demonstrate that during coinfection of HF with evolved and parental RhCMV allowed enhanced growth of the latter (as indicated by genomic DNA qPCR using strain-specific primers).

2. To demonstrate the specificity of the PKR evasion phenotype as functionally important to enhanced growth on HF (as opposed to evasion of other HF-specific factors), the virus strain that was evolved on PKR-deficient HF should be used as a control in virus replication experiments as in Fig. 2C.

3. It is important to quantify the specific increase of TRS1 expression in the evolved relative to the parental viruses and perhaps even examine the correlation of this with the degree of growth enhancement. Ideally this would be done by qRT-PCR but at a minimum could be done by image pixel quantitation of Fig. 7.

4. The authors should explain why 3 bands are evident in the TRS1 immunoblot image in Fig. 7.

**Part III – Minor Issues: Editorial and Data Presentation Modifications**

Reviewer #1: None.

Reviewer #2: In several places the authors mention increases in virus fitness, but this is narrowly defined in reference to human fibroblast infections in the presence of PKR, whereas the accompanying deletions lead to potentially worse replication cells in lacking PKR and also likely in natural animal infections given details of the virus genes deleted in cell-adapted isolates. It might be easiest to modify the language to be more precise about gains in virus replication versus the more loaded term “fitness”. Related to this point, how well do EV1, EV2, and EV3 viruses replicate in rhesus fibroblasts?

I couldn’t find any details about the serial infection protocols for experimental evolution besides the general schematic in Figure 2. Important details seem to be missing from the legend and/or methods section.

Consider showing actual datapoints instead of bar graphs in Figures 1 and 2.

How much read depth was obtained on the parental stocks to determine if the TRS1 duplication events are already present in rare genomes?

Would the combinations of deep sequencing and Sanger based analysis reveal if there are some virus genomes replicating with more than 2 copies of rTRS1?

To more definitively link the Pac sequences to the recombination process it would be ideal to have a virus strain with altered or deleted Pac sites to test whether they promote recombination under selection of human cells. Such experiments are not required in a revision for this reviewer, but might be mentioned as a future direction if the authors agree that this is a feasible and useful experiment.

I would consider modifying the title of the manuscript to highlight the isomerization of RhCMV to resemble hCMV through a single inverted duplication event.

Reviewer #3: See above

PLOS authors have the option to publish the peer review history of their article (what does this mean?). If published, this will include your full peer review and any attached files.

Reviewer #1: No

Reviewer #2: No

Reviewer #3: No
---

## [Editor Report · Decision Letter 1]

4 Jan 2021

Dear Dr. Geballe,

We are pleased to inform you that your manuscript 'Rapid adaptation to human protein kinase R by a unique genomic rearrangement in rhesus cytomegalovirus.' has been provisionally accepted for publication in PLOS Pathogens.

Best regards,

Robert F. Kalejta

Associate Editor

PLOS Pathogens

Blossom Damania

Section Editor

PLOS Pathogens

Kasturi Haldar

Editor-in-Chief

PLOS Pathogens

orcid.org/0000-0001-5065-158X

Michael Malim

Editor-in-Chief

PLOS Pathogens

orcid.org/0000-0002-7699-2064
---

## [Editor Report · Acceptance letter]

22 Jan 2021

Dear Dr. Geballe,

We are delighted to inform you that your manuscript, "Rapid adaptation to human protein kinase R by a unique genomic rearrangement in rhesus cytomegalovirus.," has been formally accepted for publication in PLOS Pathogens.

Best regards,

Kasturi Haldar

Editor-in-Chief

PLOS Pathogens

orcid.org/0000-0001-5065-158X

Michael Malim

Editor-in-Chief

PLOS Pathogens

orcid.org/0000-0002-7699-2064